# Infectious Complications in Patients with Myelodysplastic Syndromes: A Report from the Düsseldorf MDS Registry

**DOI:** 10.3390/cancers16040808

**Published:** 2024-02-16

**Authors:** Annika Kasprzak, Julia Andresen, Kathrin Nachtkamp, Andrea Kündgen, Felicitas Schulz, Corinna Strupp, Guido Kobbe, Colin MacKenzie, Jörg Timm, Sascha Dietrich, Norbert Gattermann, Ulrich Germing

**Affiliations:** 1Department of Hematology, Oncology and Clinical Immunology, Heinrich-Heine University, 40225 Duesseldorf, Germanykuendgen@med.uni-duesseldorf.de (A.K.); kobbe@med.uni-duesseldorf.de (G.K.); gattermann@med.uni-duesseldorf.de (N.G.); germing@med.uni-duesseldorf.de (U.G.); 2Institute of Medical Microbiology and Hospital Hygiene, University Hospital Duesseldorf, 40225 Duesseldorf, Germany; 3Institute of Virology, Medical Faculty, Heinrich-Heine-University, 40225 Düsseldorf, Germany

**Keywords:** myelodysplastic syndromes, infectious complications, prognosis

## Abstract

**Simple Summary:**

In patients with hematologic malignancies, especially those undergoing intensive treatments like chemotherapy, infections pose a significant and life-threatening risk. This is particularly true for individuals with myelodysplastic syndromes (MDS), a condition commonly affecting the elderly and characterized by low blood-cell counts, including anemia and neutropenia. A retrospective study involving 1593 patients from the Düsseldorf MDS Registry aimed to address two key objectives: describe the incidence of infections in MDS patients and identify risk factors contributing to increased susceptibility to infections. The study highlights the critical need for tailored approaches to prevent and manage infections in immunocompromised individuals, underlining the importance of understanding and addressing factors influencing the risk of developing infections in this vulnerable patient population.

**Abstract:**

Despite notable advancements in infection prevention and treatment, individuals with hematologic malignancies still face the persistent threat of frequent and life-threatening complications. Those undergoing chemotherapy or other disease-modifying therapies are particularly vulnerable to developing infectious complications, increasing the risk of mortality. Myelodysplastic syndromes (MDS) predominantly affect the elderly, with the incidence rising with age and peaking at around 70 years. Patients with MDS commonly present with unexplained low blood-cell counts, primarily anemia, and often experience varying degrees of neutropenia as the disease progresses. In our subsequent retrospective study involving 1593 patients from the Düsseldorf MDS Registry, we aimed at outlining the incidence of infections in MDS patients and identifying factors contributing to heightened susceptibility to infectious complications in this population.

## 1. Introduction

Despite significant progress in preventing and treating infections, there remains a frequent and life-threatening complication in individuals with hematologic malignancies [1,2,3]. Patients undergoing chemotherapy or other disease-modifying treatments are particularly susceptible to developing infectious complications, which can lead to increased mortality rates. Myelodysplastic syndromes (MDSs) primarily affect the elderly population and the incidence of this disease increases with age, peaking at around 70 years [4,5]. Patients with MDS typically present with unexplained low blood-cell counts. While anemia is the most common cytopenia observed, a majority of patients also experience varying degrees of neutropenia during the progression of their illness.

### Infectious Complications in MDS

In MDS, dysplastic hematopoiesis leads to quantitative as well as qualitative [6] defects in neutrophils [7], predisposing patients to infections. Additionally, therapies inducing myelosuppression, especially treatment options for higher-risk patients, may transiently even worsen pre-existent neutropenia.

Goldberg et al. conducted an important study that investigated infections in a population of individuals with MDS. They utilized data from the US Medicare database to compare a large group of MDS patients with a non-MDS population in terms of the occurrence of infectious complications over their lifetimes. The study found that MDS patients in the cohort experienced significantly higher rates of infectious events compared to the control group. Pneumonia, sepsis, and bacteremia were identified as the most commonly observed infections [8].

Another retrospective survey, conducted at the MD Anderson Cancer Centre, focused on determining the causes of death in MDS patients. The study specifically examined patients with low and intermediate-risk MDS based on the IPSS classification. The results revealed that 84% of deaths in this patient population were directly related to MDS. Among these deaths, infectious complications emerged as the most prevalent cause. Previous works confirmed these findings [4,9].

In a large-scale retrospective study conducted by Nachtkamp et al. using data from the Düsseldorf MDS Registry, it was confirmed that infections were the leading cause of death in MDS patients. The study further demonstrated that the incidence of disease-related deaths increased as neutrophil counts decreased and were more common in patients with higher-risk disease categories [1].

Additionally, a study by Weber et al. delved into the relationship between iron levels and infections in cancer patients, providing insights that could be relevant to understanding infectious complications in MDS patients relating to other prognostic parameters [10].

Most of the data obtained up to now stem either from clinical trials representing a selected patient population or are of retrospective nature. Analyses on infections during clinical trials are biased by strict patient eligibility criteria and the administered therapies, hence therefore not reflecting infectious complications and their hazards in a real-life setting. Furthermore, infectious episodes are no primary endpoint since the influence of infections on overall survival is not investigated in this context.

## 2. Methods

For our retrospective analysis, we obtained the follow-up data of 1593 patients diagnosed with classical MDS from our Düsseldorf MDS registry and with at least one documented episode of infection. MDS was diagnosed in these patients between the years 1988 and 2019. Infections were defined as clinical symptoms of infection requiring antibiotic therapy, the isolation of a pathogen, or identification of an infection site through physical examination. The infectious episodes were further categorized as fever of unknown origin (body temperature with no identified cause after three days), microbiologically documented infection (microbiological evidence of infection), or clinically documented infection (clinically proven infection without microbiological evidence). The diagnosis and classification of MDSs were performed according to the WHO 2016 classification of myeloid neoplasms. To compare our study cohort, we also included a control cohort of 2194 patients from our registry who did not experience any infectious complications throughout the course of their disease.

Categorical parameter frequencies were presented through cross-tabulation, and distinctions were determined using Student’s *t*-test. The Mann–Whitney U test was employed for evaluating differences in continuous variables. Overall survival for the complete cohort was computed from the initial diagnosis to either death from any cause or the last follow-up date. The Kaplan–Meier methodology was utilized to construct time-to-event curves, accompanied by log-rank tests for univariate analyses. Univariate analyses were conducted with the Cox regression model. A *p*-value < 0.05 was regarded as statistically significant across all analyses. IBM SPSS (SPSS Inc., Chicago, IL, USA) and GraphPad Prism 9 (GraphPad Software Inc., La Jolla, CA, USA) were the tools employed for statistical analyses.

## 3. Results

### 3.1. Type of Infections and Most Common Sites

Our study cohort of 1593 patients included MDS patients with at least one documented infectious complication during the course of their disease. The observation period during which infections were documented was six years, starting from the time of initial diagnosis in the individual patient. The median age at diagnosis was 66 years, with a range of 18 to 99 years. Within the cohort, there were 655 females (41.4%) and 938 males (58.9%). The distribution of MDS subtypes, based on the WHO classification of myeloid neoplasms, is presented in Table 1.

The majority of infections in our study (*n* = 1312, 82.3%) were attributed to bacterial pathogens. In 154 patients (9.9%), the origin of the infectious complications remained unknown. A smaller number of patients experienced viral infections of any site (44 cases, 2.6%) and fungal infections (83 cases, 5.2%). Specifically, the bacterial causes of infections that we were able to identify were consistent with those commonly associated with febrile neutropenia, such as enterobacteria and coagulase-negative staphylococci. Table 2 provides an overview of the most frequently detected pathogens identified through standard blood cultures.

Among the identified infectious complications, pneumonia was the most frequent site of infection, accounting for 439 cases (27.5%). This was followed by fever of unknown origin, observed in 394 cases (24.7%). Other common sites of infection included bloodstream infections (154 cases, 9.6%), urinary tract infections (152 cases, 9.5%), and sepsis (128 cases, 8.0%). Furthermore, 243 patients (15.3%) experienced complications related to the insertion of central or peripheral venous catheters.

### 3.2. The Impact of Infections on Overall Survival

During the period spanning from 1988 to 2019, infectious complications emerged as the leading cause of death in our study cohort, constituting 32% (*n* = 512) of the total deaths. The next most common causes of death were transformation into AML, accounting for 16% (*n* = 256) of the deaths, and hemorrhage, which contributed to 5.7% (*n* = 91) of the fatalities.

In our study, we compared the overall survival (OS) of our study cohort, which included MDS patients with infectious complications, with a control group from the Düsseldorf MDS Registry consisting of MDS patients who did not experience any infectious complications during their disease course (Figure 1). The analysis revealed a significant difference in median OS between the two groups. The control group without infections had a superior median OS of 37 months (95%CI 33.5–40.4), while patients with at least one infectious complication had a shorter median OS of 21 months (95%CI 18.9–23.1) (*p* < 0.001).

Furthermore, we examined the impact of different types of infectious complications on survival within the study cohort. Patients who experienced viral infections had the longest median OS of 38 months (95%CI 21.9–54.1) among those with infectious complications. Patients with bacterial and fungal infections had median OS values of 22 months (95%CI 19.5–24.5) and 21 months (95%CI 11.3–30.7), respectively. Patients in whom no pathogen could be detected, meaning fever of unknown origin or clinically documented infection, had the shortest median OS of 15 months (95%CI 9.6–20.4). The differences in OS among these four groups were statistically significant (*p* < 0.001).

In our study, we observed that the majority of patients (*n* = 484, 89%) experienced between two and three bacterial infections over a period of six years. Only a small number of patients had only one or more than three bacterial infections. However, when comparing the overall survival of patients based on the number of infections, we did not find a statistically significant difference (*p* = 0.172, 95%CI 20.3–27.7).

### 3.3. Patient-Related Parameter

Age at the time of initial diagnosis is an important non-disease-related factor which plays a crucial role in prognostic stratification and therapeutic decision-making for MDS patients. The median age of our study group at initial diagnosis was 66 years (range 18–99 years). In univariate analyses comparing patients who had no infections versus those who had one or more, patients with older ages (>70 years) tended to have a higher incidence of infectious episodes (*p* < 0.001). In addition, patients older than 70 years suffering from infections had a shorter survival than patients who did not experience an infectious complication. The difference in OS was highly significant, with 14 months (95%CI 11.9–16.0) for patients suffering from infectious complications compared to 24 months (95%CI 20.6–27.4, *p* < 0.001) (Figure 2).

To estimate the difference in infectious incidences between MDS patients and a non-MDS population, we chose to examine the incidence of pneumonias as an example. The incidence of pneumonias in Germany in residents older than 65 years of age was 1197 per 100,000 in 2017. Considering patients older than 65 years and suffering from pneumonia in our study cohort, the incidence was 6.9 times higher.

### 3.4. Disease-Related Parameter

#### The Role of Peripheral Blood Counts

While 608 patients in our cohort had lower-risk disease (77.4%), 178 patients (22.6%) suffered from high-risk MDS (IPSS-R > 5 points). Lower risk disease included patients with low risk and intermediate risk (IPSS-R < 1.5 points to 4.5 points), as well. When comparing the overall survival of those two groups, we could not ascertain a difference in median survival (*p* = 0.318, 95%CI 24.7–31.2).

In total, 70.2% of the patients (*n* = 1119) presented with cytopenia of at least one or more hematopoietic lineage. While thrombocytopenia defined as <100,000 thrombocytes/µL was the most frequent (*n* = 674, 42.3%), neutropenia, defined as an ANC of <1800 neutrophils/µL, as well as anemia with a hemoglobin level as low as 8 g/dL, occurred in 298 and 662 of the patients (18.7% and 41.5%).

We found a significant negative correlation between the severity of anemia and the incidence of infectious complications, meaning that patients suffering from severe anemia at time of diagnosis are more likely to develop infections during the course of the disease (*p* < 0.001). It is essential to mention that patients who were anemic with a hemoglobin of less than 8 g/dL proved to be significantly more often neutropenic with less than 800 neutrophils/µL (*p* = 0.010). This finding was independent from lower- or higher-risk disease according to the IPSS-R. As a hemoglobin threshold of 10 g/dL is recommended by the IPSS-R to distinguish mild from severe anemia in daily practice, we investigated overall survival regarding degree of anemia. Severely anemic patients had a significantly inferior overall survival compared to patients only presenting with mild anemia of 14 months (95%CI 12.3–15.7) and 29 months (95%CI 24.9–33.0), respectively (*p* < 0.001). 

Concerning the absolute neutrophil count (ANC), we were able to confirm a significant negative correlation between the degree of neutropenia and infections, as well (*p* < 0.001). As a threshold of 800 ANC/µL is recommended to define aplasia according to the IPSS-R, we chose this value for our analyses. Absolute neutrophil counts of as low as 800/µL were related to inferior overall survival of 14 months (95%CI 11.2–16.8), compared to 24 months (95%CI 20.9–27.1) in non-neutropenic patients (*p* = 0.002). When deploying a cut-off of 1800/µL, the difference in OS was comparable with 14 months in neutropenic patients (95%CI 11.1–16.8) and 21 months (95%CI 17.2–24.7) in non-neutropenic patients.

No correlation could be found between thrombocytopenic patients and numbers of infectious episodes (*p* = 0.241). 

### 3.5. The Role of Bone Marrow Blasts

To determine the impact of blast percentage in the bone marrow on survival in patients with infectious complications, we divided our study group into patients with more or less than 5% blasts in the marrow. Blast percentage had an excruciating impact on median OS in our patients: 798 patients with <5% blasts had a superior prognosis than 682 patients with more than 5% blasts, with 30 months (95%CI 26.1–33.9) compared to 13 months (95%CI 11.2–14.8), respectively (*p* < 0.001, Figure 3).

### 3.6. Multivariate Analysis for Disease-Related Factors

In our study, we conducted a multivariate Cox proportional hazards model, incorporating median OS as a time-varying covariate. Additionally, we included various disease-related parameters as categorical covariates, specifically peripheral blood-cell counts such as hemoglobin, ANC, WBC, monocytes, lymphocytes, and platelets. The objective was to determine the impact of these parameters on OS in patients who experienced infectious incidents.

Our analysis revealed four parameters that were significantly associated with poorer OS in patients with infectious complications. These parameters were as follows: hemoglobin (Hb) levels below 9 g/dL (*p* < 0.001), WBC count below 4000/µL (*p* = 0.038), ANC below 800/µL (*p* < 0.001), and platelet count below 50,000/µL (*p* = 0.003). These findings indicate that lower levels of Hb, WBC, ANC, and platelets are predictive of a reduced OS in MDS patients who experienced infectious incidents.

In addition to examining the impact of these variables on overall survival, we also conducted a multivariate linear regression analysis to investigate their influence on the probability of developing infectious complications. Among the variables studied, the most significant negative prognostic marker for developing infections was an ANC below 800/µL, as shown in Table 3. On the other hand, higher hemoglobin levels (>9 g/dL) were associated with a reduced risk of experiencing infectious episodes. These findings suggest that maintaining adequate ANC levels and higher hemoglobin levels may play a protective role in reducing the risk of developing infectious complications in patients with MDS.

### 3.7. The Influence of Therapy on Infectious Susceptibility

In the present study, 66% of the patients undergoing therapy with hypomethylating agents (HMA) developed an infectious episode. Patients receiving best supportive care or therapy with lenalidomide suffered from fewer infections, with 36% and 32%, respectively. The difference in infectious incidents between these subgroups was statistically relevant (*p* < 0.001).

In total, 129 patients received best supportive care alone. A total of 82 patients (64%) suffered from an infectious complication, while 47 (36%) did not suffer from infections. We could not ascertain a difference in survival between these two subgroups, with 24 months (95% 20.4–27.8) and 25 months (95%CI 21.5–31.6), respectively (*p* = 0.070).

In our study cohort, 138 patients presented with iron overload (ferritin levels exceeding 1000 µg/L) at initial diagnosis. Patients with iron overload had significantly more infectious episodes than patients with normal ferritin levels (*p* < 0.001). Amongst these patients, 17 (12%) received iron chelation. Those treated with iron chelators gained a superior survival with 121 months compared to 23 months in patients not receiving chelation (*p* < 0.001, 95%CI 21.8–34.1). Except for the other best supportive care measures, patients administered chelation did not undergo disease-altering treatment. 

In lower-risk MDS, Lenalidomide is approved for patients presenting with transfusion-dependent anemia and del(5q) by the EMA and FDA. In our cohort, 36 patients were treated with Lenalidomide and suffered an infectious complication of any type during their disease. However, patients who received Lenalidomide and who suffered from infections had no shorter overall survival than patients under Lenalidomide therapy without infections (*p* = 0.933, 95%CI 23.8–36.1).

For advanced MDS, HMAs are frequently used either as a monotherapy for patients who are not eligible for allogeneic stem-cell transplantation or as a means of bridging to allogeneic stem-cell transplantation to reduce the blast count. A total of 295 patients received HMAs; amongst them, 17 patients (5.7%) received HMAs as a bridging to transplant. In total, 95 patients (32%) in this subgroup passed away. The majority of deaths were disease-related, with 35 deaths (37%) due to infections and 30 (32%) due to transformation into AML. Patients treated with HMAs and suffering from infectious episodes had an inferior median overall survival of 26 months compared to patients who did not suffer infections and received HMA treatment (49 months, *p* = 0.011, 95%CI 29.8–44.1). When excluding patients undergoing alloSCT after receiving HMAs, the survival difference between patients suffering from infections and those without is even more pronounced (24 months vs. 43 months, *p* = 0.005, 95%CI 28.5–39.4). 

### 3.8. Therapy of Infectious Complications

The majority of patients (*n* = 1045, 65.7%) suffering from infectious complications received anti-infectious therapy. Amongst them, 948 patients were treated with antibiotics, while 207 patients were in need of antimycotics. Patients who received any kind of anti-infectious therapy had a median survival of 22 months (95%CI 19.1–24.8). Compared to patients who did not receive any kind of therapy, their survival was statistically shorter (20 months, 95%CI 16.9–23.9, *p* = 0.021, Figure 4). The reasons for why no antimicrobial therapy was administered are unknown. Of the 1593 infectious events, 258 (16%) required hospitalization. Unfortunately, we were not able to ascertain the number of patients receiving prophylactic treatment. 

## 4. Discussion

Up to now, there is no standard protocol on the treatment and prophylaxis of infectious episodes in MDS patients [11,12,13]. Since patients suffering from MDS tend to be older and harbor more comorbidities compared to other patients with hematological malignancies, they belong to a patient group which is particularly susceptible to infectious complications. Moreover, MDS patients are a heterogeneous population in terms of MDS subtypes, prognosis, treatment approaches, and patient-specific risk factors, such as performance status and comorbidities. Given these complexities, we conducted a study utilizing a large cohort of patients with documented infectious incidents to investigate the influence of patient-related and disease-related parameters on overall survival in MDS patients who experienced infectious complications. By analyzing these factors, we aimed to gain insights into the impact of various parameters on the OS of MDS patients with infectious complications.

Within the scope of our work, we identified a representative study population from our Düsseldorf MDS Registry who developed at least one infectious episode during their disease. Age emerged as an important contributing factor to the development of infections, with older patients at the time of initial diagnosis being more susceptible to infectious complications.

MDS patients demonstrated a heightened vulnerability to bacterial infections, particularly when they experienced cytopenia. The severity of cytopenia in any of the hematopoietic lineages was correlated with an increased likelihood of developing infectious episodes. Among the infections documented, pneumonia constituted the largest proportion, followed by fevers of unknown origin and septicemia.

The most frequently detected pathogens in our study were Escherichia coli, Staphylococcus aureus, and coagulase-negative staphylococci. The identification of these commonly occurring pathogens aids in the understanding of the microbial landscape in MDS-associated infections.

Older age proved to be a negative prognostic factor with regard to OS. Although not statistically significant in our cohort, several studies investigated the impact of comorbidities and found an important association, especially in patients with cardiovascular diseases [14]. Most importantly, MDS patients developed more infectious episodes than patients of the same age in a non-MDS population. This finding was especially pronounced in patients older than 70 years of age.

Our findings revealed that ANCs below both 800/µL and 1800/µL were associated with overall survival in the context of infectious episodes. Moreover, the presence of neutropenia significantly increased the risk of developing an infection and emerged as the strongest predictor in our multivariate analyses. Neutropenia is a common occurrence in newly diagnosed MDS patients, with approximately 50% of patients experiencing this condition [15]. Its prevalence is even higher in higher-risk MDS cases. In lower-risk and early-stage MDS, neutropenia is likely associated with the apoptosis of hematopoietic progenitor cells. In advanced stages of MDS, particularly in higher-risk cases, neutropenia is often a consequence of generalized hematopoietic insufficiency and progressive leukemic proliferation [16]. At the cellular level, neutrophil impairment and dysfunctional monocytes can lead to an impaired immune system, making patients more susceptible to infectious episodes [17]. This immune dysfunction may contribute to the increased incidence of infections in MDS. Importantly, our study found that neutropenia was a common occurrence across various MDS subtypes, indicating its relevance in the context of infectious complications regardless of the specific subtype. These findings emphasize the importance of monitoring neutrophil counts and addressing neutropenic states in MDS patients to mitigate the risk of infections and improve overall survival outcomes.

In our study, we identified additional disease-related factors such as the percentage of bone marrow blasts at the time of initial diagnosis as significant factors influencing the prognosis of MDS patients with infectious complications. These findings highlight the importance of considering these factors in the assessment and management of patients with MDS.

Concerning the therapeutic categories we examined, we could only ascertain a survival difference in the category of patients receiving iron chelators as well as hypomethylating agents. Patients undergoing disease-altering treatment in the form of HMAs had an inferior OS. In clinical practice, disease-altering therapeutic options, like HMAs, are administered according to the patients’ performance status and IPSS-R [18]. Due to a myelosuppressive effect, certain therapies may transiently worsen pre-existing cytopenia and therefore increase the risk of infection until hematologic improvement is observed. However, the risk of developing infectious events while receiving certain drugs is poorly documented and data is highly heterogenous. Demethylating agents like azacitidine and decitabine are employed as second-line therapy in patients with low or intermediate risk, failing first-line treatment with erythropoiesis-stimulating agents and high transfusion burden [19]. High-risk patients receive HMAs as a first-line treatment, when allogeneic stem-cell transplantation is not feasible due to comorbidities or older age [20,21]. Up to now, patients receiving HMA therapy did not suffer from more or different types of infections in general. Grade 3–4 neutropenia was found to be the most relevant side effect predisposing patients to infections. However, Fenaux et al. identified higher blast percentage as a negative prognostic factor for infectious episodes. Infectious complications were especially pronounced in patients treated with azacitidine and with more than 20% blasts in the bone marrow. Furthermore, infectious incidence varied according to azacitidine dose. Patients receiving doses of 75 mg/m^2^ over the course of seven days suffered from more infectious complications than patients receiving the same dose for only five days [22]. Studies such as the AZA001-trial, evaluating the efficacy of treatment with azacitidine, found no significant difference regarding infectious incidents in patients treated with azacitidine compared to BSC, low-dose cytarabine, and induction chemotherapy [23]. In this present study, patients undergoing HMA treatment suffered the most infectious episodes compared to other therapeutic categories. Studies showed that azacitidine developed a pronounced myelosuppressive effect especially during the first cycles. Common hematologic events were anemia, thrombocytopenia, and neutropenia [24]. Various studies reported adverse events labeled as Grade 3 or 4 concerning neutropenia in 80% up to 90% of patients. In a large retrospective study, Merkel et al. [25] found ANC prior to each azacitidine cycle to be an important risk factor in univariable analysis but it did not maintain its prognostic relevance in a multivariate model. The most relevant prognostic factors in multivariable analysis were low platelet counts, followed by poor risk cytogenetics, and low hemoglobin levels (<10 g/dL). The same group reported a higher risk for infectious complications in patients who were treated with azacitidine 75 mg/m^2^ for seven days than those who received only five days of treatment within the first cycle [25].

In our study, we found that MDS patients who received anti-infectious therapy, specifically antibiotics and antifungals, had significantly better OS. This suggests that the timely and appropriate treatment of infectious complications can have a positive impact on patient outcomes. It is important to note, however, that only a small proportion of patients required in-patient care, indicating that the majority of infectious episodes could be managed in an outpatient setting.

Our study cohort represented a large, unselected MDS population. Compared to the restricted populations in clinical trials due to strict eligibility criteria, we were therefore able to investigate a real-world MDS cohort. We were able to document infectious episodes six years since initial diagnosis in each patient, the various pathogens, and the anti-infectious therapies in a large cohort.

However, because of its retrospective nature, we were not able to retrieve complete datasets for each patient. It is therefore possible that we missed out on important inpatient data and follow-up dates. Furthermore, the parameters we deployed, i.e., the peripheral blood-cell counts and comorbidities were static variables collected at first diagnosis, meaning we were yet not able to study the dynamic impact of these parameters over time. Dynamic parameters should therefore be collected prospectively.

Unfortunately, there are no guidelines for the treatment and prophylaxis of infectious complications in MDS patients. Since patients with MDS are prone to suffer from infectious complications, especially neutropenic patients with an absolute neutrophile count of <0.8 × 10^9^/L, they should be observed closely and should receive broad-spectrum empirical antibacterials in the case of fever. Relevant factors predisposing patients to infectious complications are advanced disease with higher blast percentage, poor performance status, and older age. Treatment should be adjusted according to the above-mentioned individual risk factors and therapy [26]. Patients who receive the best supportive care measures should be monitored closely for neutropenia [27]. However, the mere presence of neutropenia does not justify the use of anti-infectious treatment since long-term treatment with antibiotics may induce resistances and adverse effects [28].

In conclusion, our study has provided valuable insights into the impact of infectious complications on the overall survival of patients with MDS. The presence of cytopenia, poor performance status, and advanced age were identified as significant negative prognostic markers in relation to infectious incidents. To improve the management of infections in MDS patients, further prospective research is needed. It is crucial to establish a standard protocol for the prevention and treatment of infections specific to the MDS population. Large-scale, prospective trials are necessary to develop consensus guidelines for antibacterial and growth-factor prophylaxis in MDS. These trials should assess the benefits of prophylaxis in terms of infection prevention, infection-related mortality, overall survival, and cost-effectiveness.

By addressing these research gaps, we can enhance our understanding of infectious complications in MDS and develop evidence-based strategies to improve patient outcomes. Collaborative efforts among researchers, clinicians, and healthcare providers are essential to advance the field and optimize the management of infections in MDS patients.

## 5. Conclusions

Vigilance is required in neutropenic patients who present as unwell, hypotensive, or febrile. Especially in elderly patients, symptoms of infection may be minimal, since those patients rather present with confusional state. Prior to the administration of anti-infectious therapy, the clinical history should be checked for past infectious complications including positive blood cultures and the presence of possible antibiotic-resistant bacteria. Patients presenting with comorbidities which may increase their risk of infections or patients who suffered a serious infection during previous treatment options deserve specific alertness. Antibacterial prophylaxis may be considered in these groups of patients, especially the prophylactic use of quinolones [29,30]. However, the duration of prophylaxis when administered should be as short as possible, covering only the nadir time of the risk. These findings emphasize the importance of the early recognition and prompt treatment of infections in MDS patients. The use of appropriate antimicrobial agents tailored to the specific pathogens involved can help to improve outcomes and potentially reduce the risk of complications. It is crucial for healthcare providers to closely monitor MDS patients for signs of infection and initiate appropriate therapy promptly to optimize patient outcomes.

## Figures and Tables

**Figure 1 cancers-16-00808-f001:**
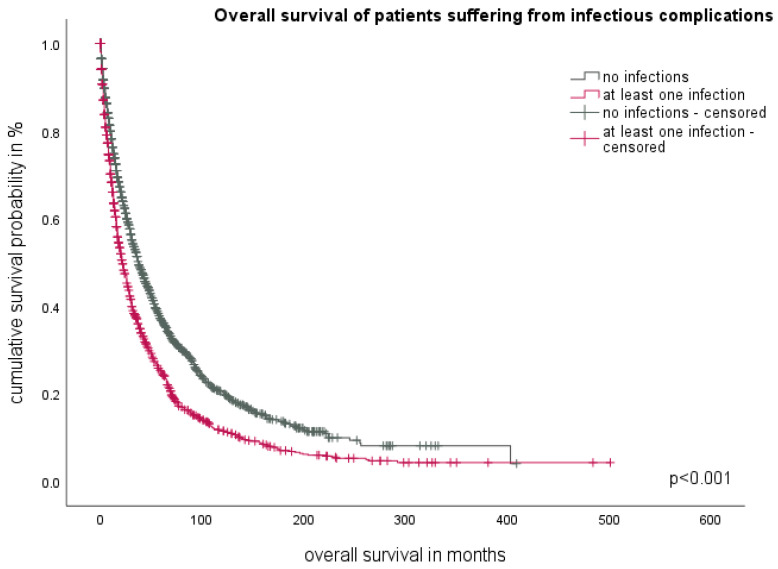
Comparison of overall survival of patients with at least one infectious episode in the study cohort vs. no infectious episode in the control cohort.

**Figure 2 cancers-16-00808-f002:**
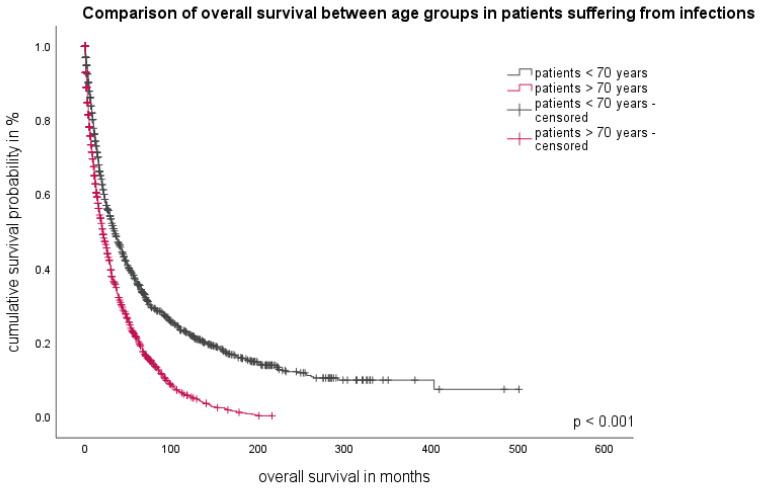
Overall survival of patients suffering from infectious complications. Comparison between patients younger and older than 70 years at time of diagnosis.

**Figure 3 cancers-16-00808-f003:**
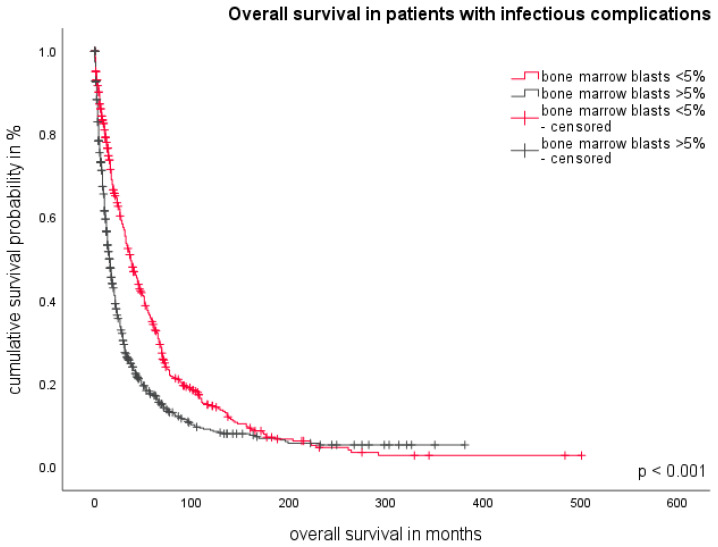
Overall survival of patients suffering from infectious complications. Comparison between >5% and <5% bone marrow blasts.

**Figure 4 cancers-16-00808-f004:**
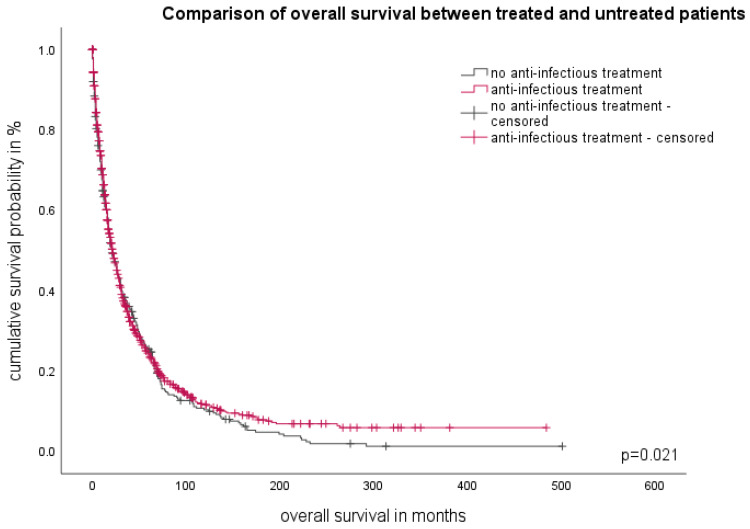
Overall survival of patients according to anti-infectious treatment. Comparison of treated and untreated patients.

**Table 1 cancers-16-00808-t001:** Distribution of MDS subtypes in patients suffering from infectious complications according to the WHO 2016 classification, including RAEB-T, on myeloid neoplasms.

MDS Subtype WHO 2016	*n*	%
MDS-SLD	54	3.4
MDS-MLD	446	28.0
MDS-RS-SLD	79	5.0
MDS-RS-MLD	9	0.6
MDS del(5q)	24	1.5
MDS EB 1	212	13.3
MDS EB 2	305	19.1
CMML 1	148	9.3
CMML 2	50	3.1
RAEB-T	220	13.8
MDS/MPN U	14	0.9
RARS-t	22	1.4
unclassifiable	10	0.6
total	1593	100

**Table 2 cancers-16-00808-t002:** The most common pathogens documented by positive blood cultures.

Pathogen	Number of Patients with Positive Microbiological Cultures	%
*Aspergillus*	6	0.9
*Candida albicans*	5	0.7
*Clostridium difficile*	6	0.9
*Escherichia coli*	55	8.6
*Enterococcus*	8	1.2
*Enterococcus faecium*	5	0.7
*Heamophilus influenzae*	4	0.6
*Klebsiella pneumoniae*	7	1.1
Koagulase negative streptococcus	21	3.3
*Mycobacterium tuberculosis*	5	0.7
*Pneumococcus*	7	1.1
*Proteus mirabilis*	7	1.1
*Pseudomonas aeruginosa*	17	2.6
*Staphylocuccus aureus*	22	3.4
*Staphylocuccus epidermidis*	13	2.0
*Staphylococcus*	9	1.4
*Stenotrophomonas maltophilia*	6	0.9
Others	57	8.9
No pathogen detected	378	59
total	638	100

**Table 3 cancers-16-00808-t003:** Risk of infectious episodes depending on peripheral blood counts.

Parameter	χ^2^	*p*	Hazard Ratio	95%CI
Hb > 9 g/dL	26.499	<0.001	0.675	0.58–0.78
ANC < 800/µL	45.609	<0.001	2.042	1.65–2.51
WBC < 4000/µL	0.024	0.875	1.013	0.85–1.19
Pla < 50,000/µL	0.395	0.529	1.060	0.88–1.27

## Data Availability

Data is unavailable due to privacy restrictions.

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
