# Peer review of "Infectious Complications in Patients with Myelodysplastic Syndromes: A Report from the Düsseldorf MDS Registry"

_cancers, 2024, doi:10.3390/cancers16040808_

Round 1

Reviewer 1 Report (Previous Reviewer 3)

Comments and Suggestions for Authors

No further suggestions.

Paper can be accepted in the present form.

Reviewer 2 Report (Previous Reviewer 4)

Comments and Suggestions for Authors

All issues were addressed by the authors and the paper can be published in its present form.

Reviewer 3 Report (Previous Reviewer 5)

Comments and Suggestions for Authors

Changes are adequate.

This manuscript is a resubmission of an earlier submission. The following is a list of the peer review reports and author responses from that submission.

Round 1

Reviewer 1 Report

Comments and Suggestions for Authors

This study is a retrospective analysis conducted on a very large series, in which the follow-up data of 1,593 patients diagnosed with MDS from the Düsseldorf MDS registry and who presented at least one documented episode of infection were collected and analyzed.

The objectives of this study were:

- describe the incidence of infections in patients with MDS

- identify the factors that contribute to a greater susceptibility to infectious complications

- provide insights for improving  infection management in these patients with a specific focus on neutropenia

These aspects had not been completely clarified by previous studies, cited by the Authors, in particular the identification of risk factors for infectious complications, and their influence on prognosis.

MDS was diagnosed in these patients between 1988 and 2019.

The period of time during which infectious episodes of any type have been documented is six years from the first diagnosis relating to the individual patient.

To compare the cohort of this study, a control cohort was included of 2,194 patients from the same registry who did not experience infectious complications throughout the course of their disease

The majority of infections in this study (n=1,312, 82.3%) were attributed to bacteria , as confirmed by positive microbiological results through standard blood tests. Among the infectious complications identified, pneumonia was the most frequent cause of infection (439 cases, 27.5%). This was followed by fever of unknown origin (394 cases, 24.7%). Other common sites of infection included positive blood cultures (154 cases, 9.6%), urinary tract infections (152 cases, 9.5%), and sepsis (128 cases, 8.0%). In approximately 30% of patients it was not possible to identify a specific focus of infection

During the period from 1988 to 2019, infectious complications emerged as the leading cause of death in the study cohort, constituting 32% (n=512) of total deaths.

The study evaluated the influence of patient- and disease-related parameters on overall survival in patients with MDS who experienced infectious complications. .

As regards patient-related factors, advanced age was associated with both an increased risk of infectious complications and a worse Overall Survival (OS) in patients who developed infections, while an unfavorable performance status (Karnofsky-Index  < 80%) was also associated with inferior OS in patients with infectious complications

As regards disease-related factors, both anemia and neutropenia (both < 800 and < 1,800) were associated with both an increased risk of developing infections and a worse OS.

A ferritin level > 1000 ng/mL was also associated with both a greater risk of infections and poorer OS

Multivariate analysis showed that 4 factors were associated with poorer survival in patients with infectious complications: Hb level < 9 g/dL,  leukocyte count < 4,000,  neutrophil count < 800, and  platelet count < 50,000

As for the risk of developing infectious complications, a neutrophil count < 800 was associated with a greater risk, while an Hb level > 9g/dL showed a protective role.

 As for therapy-related factors, patients who received azacitidine at the standard dosage (75 mg/m2 for 7 days) showed more infectious complications than patients who received a reduced dosage (the same dose for only five days)

Furthermore, MDS patients who received anti-infective therapy (antibiotics and antifungals) had significantly better OS.

As for the limitations of this study, as the Authors correctly point out, due to its retrospective nature, it was not possible to recover complete data for a proportion of patients. It is therefore possible that important data on hospitalized patients and subsequent updates were not recorded. Furthermore, it was possible to evaluate the Karnofsky index only in 538 patients (of which 266 with a performance status <80%). Finally, the parameters used, the peripheral blood cell count and comorbidities, were static variables collected at the first diagnosis , and therefore it was not possible to study the dynamic impact of these parameters over time. This may also explain why the impact of comorbidities was not statistically significant in this patient cohort, contrary to the results of other studies (especially regarding cardiovascular diseases).

The results of this study might be the starting point for future prospective studies, aiming at  improving the prevention and treatment of infectious complications in patients with MDS.

Despite these limitations, this study provides many important information which, even if intuitable by clinicians, were confirmed on a very large and unselected cohort of MDS patients from the real world. In particular, it was possible to document the infectious episodes six years after the initial diagnosis in each patient, the various pathogens and the anti-infective therapies that have been administered.

For this reasons, in my opinion, the manuscript deserves publication.

Author Response

Thank you for your detailed feedback. 

Reviewer 2 Report

Comments and Suggestions for Authors

Kasprzak A et. al. investigated the risk of infectious complications in MDS patients by using their large patient cohort: Dusseldorf MDS Registry and showed some results.

  1. Infection experience increased the shortage of overall survival but the infection times are not associated.
  2. Older age patients (>70 years) have a significantly higher incidence of infections and it shortages OS.
  3. Men have more infections than women.
  4. MDS patients have 6.9 times higher incidents pneumonia compared to non-MDS >65 ages.  
  5. In peripheral blood counts,  low ANC is associated with infectious complications but not WBC <4000/ul and Platelet <50000/ul. Moreover, high Hemoglobin (>9g/dl) reduces the risk of infectious complications. However, iron overload is the risk of infections though that of OS is improved than normal iron levels.
  6. HMA treatment is not associated with infected death but the pretreatment of HMA in transplant improved OS.
  7. Therapy of infection improved OS.

The results are interesting and important data to thinking about MDS of infectious complications, there is some revised points to improve the article as shown below;

  1. The author should reconsider the title and abstract, background for a better summary of their results. 

  1. Figure 1 should be referenced in line 130.

  1. It was not clear how Karnofsky-Score and blast percentage are associated with MDS patients of infectious complications.

  1. The author should combine peripheral blood count sections line 185  and 251.

  1. Line 304, “Patients who received any kind of anti infectious therapy had a median survival of 22 months (95%CI 19.1-24.8). Compared to patients who did not receive any kind of therapy, their survival was statistically longer (20 months, 95%CI 16.9-23.9, p=.021).” It should be wrong written and to revise “their survival was statistically shorter”?

  1. Figure 6 should be referenced in line 307.

  1. For discussion, the current one is too long, and reconsider the contents due to the result (for example,  bone marrow microenvironment is not mentioned in the result section).

Author Response

We sincerely appreciate your valuable feedback, and we believe that these revisions will significantly enhance the quality and coherence of our manuscript. Please find our detailed responses in the following.

Reviewer 3 Report

Comments and Suggestions for Authors

The present deals with a relevant issue and analize a large amount of patients. However there are several flows that need extensive correction.

75-76 Please define:  1.“fever of unknown origin “, 2.“microbiologically documented infection” and 3. “clinically documented infection” (and use the order 2. – 3. – 4.)

93-95: please clarify the sentence : “The time period …. The individual patients”; what doas “six years” means, exactly ?

101-109: please rewrite and classify infetious episode as: 1.“fever of unknown origin “, 2.“microbiologically documented infection” and 3. “clinically documented infection”; devide “microbiologically documented infection” and “clinically documented infection” as per site of infection (eg pneumonia, UTI, etc.) and patogens.

Table 2: authors reported onliy blood cultures; are thera cultures from different source (eg urine, nasal swab, sputum ?)

135-136: “Patients in whom no pathogen could be detected”: define as either 1.“fever of unknown origin “ or 3. “clinically documented infection”  140-142: please report

160: Correct Figure number (correct = 1)

Table 3: clarify: first row, second column: “2” ?

267-268: “the patients […] developed and infectious episode “ ; what does it mean: a SECOND episode after the first episode, required for study inclusion ?

272-273: the same question of above: what does it mean: a SECOND episode after the first episode, required for study inclusion ?

325-327: please clarify: “… more than 50%... ”; where does it come from ?

356-358: bibliopraphy

369-371: bibliopraphy

412-414: bibliopraphy

Author Response

Thank you for your detailed feedback, and we will incorporate these revisions to improve the clarity and accuracy of the manuscript.

Reviewer 4 Report

Comments and Suggestions for Authors

The investigators found impact of infectious complications on overall survival for MDS patients. The presence of cytopenia, poor performance status and older age were identified as significant negative prognostic factors for MDS patients with infections.

Overall the paper is well written with significant results and potential impact on clinical practice.

I have a major issue that in the conception of the study WHO 2016 classification was used and Table 1 is misleading as CMML and AML patients (formerly RAEB-T in FAB 1982) were falsely added as they were excluded from the WHO MDS classification since 2001.

Authors should reclassify:

1.     According to WHO 2022 and ICC 2022 and compare if there is any modification of their results.

2.     It would be interesting to exam the TP53 genetic subgroups as defined by WHO 2022 and ICC (multi hit vs single hit compared to blast counts)

3.     Classify CMML in the MDS/MPN subgroup (any difference for proliferative CMML patients?)

4.     Classify RAEB-T correctly to AML

Author Response

In conclusion, we are grateful for the reviewer's insightful comments and constructive suggestions. We are committed to addressing these concerns promptly, and we believe that these adjustments will strengthen the robustness and accuracy of our study, ensuring its alignment with the latest standards in hematological classification

Reviewer 5 Report

Comments and Suggestions for Authors

Interesting study with all problems of a registry study. We do not know how reliable events are reported and therefore, firm conclusions are not possible. However, in MDS we do not have better data until now.

Specific remarks:

Main part of the abstract and the introduction are identical. Please adapt abstract to include more results of the study.

RAEB-T (220pt 13.8%)  are not part of WHO 2016 (in methods). Please clarify. In addition, the significant group of MDS/MPN are not pure MDS. Please clarify and add more details (do you see the same frequency of neutropenia, infections...). 

Ferritin as the sole parameter for definition of iron overload is not adequate. If secondary causes are exculded and Trf saturation is high (as well as multitransfusion was preformed), high ferritin suggests an iron overload. In addition, iron chelation was related to more infections. If possible report how many patients have been on iron chelation when acquiring an infection.

Data cut off is not clear (only stated, when the patients were diagnosed). If the data cut off was after the pandemia, I wonder, why there is no case of COVID. It would be of importance to know, what happened to these patients. In addition, some infulenza cases should have occured in such a large population (in contrast to quite detailed information about bacterial infections).

The use of antiinfective treatment in these patients should be standard in case of infections. However, it would be of interest, how many where on prophylactic treatment and how many had a treatment of the infection itself. 

In clinical practice, neutrophils <500G/l are known to be related to a significant higher infection rate. How was the value of 800G/l determined (continous variable)? If mortality is increasing, would you encourage the MDS community to  investiagte this value in prospective trials?

The number of patients with bacterial infections is 1312. It is written, that this infections were proven by blood culture. In table 2 is written: "patients with positive microbiological cultures". The total number is 638. Of this 638 378 were negativ. These numbers and titels are not correct in the sense reported. Please explain or correct. In addition 15.3% of insertions of central/peripheral lines were complicated by infections. How high is the general rate of infections after central/peripheral line insertion. Please explain this high number.

In the discussion, you elaborate on the correlation between treatment and infection. However, in the result section, there is not much about when the infection occured (at the beginning of treatment, after failure...). Do you have such parameteres in the registry to analyze this more in detail. Please explain.

The last of your 3 statments in the abstract  "provide insights into improved management strategies for infections" is not answered. You just state, that prospective trials are missing (which is absolutely correct) to answer these important questions. However, some suggestions based on your study should be done or daily practice you encounter can be directly questioned.

Author Response

We appreciate the thorough review and constructive feedback from the reviewer, and we are committed to incorporating these suggestions to improve the clarity and robustness of our study.
